# Optogenetic dissection of transcriptional repression in a multicellular organism

Jiaxi Zhao [1,7,8,12], Nicholas C. Lammers[2,9,12], Simon Alamos[3,10,11], Yang Joon Kim[2], Gabriella Martini[4] & Hernan G. Garcia [1,2,4,5,6] ✉

Transcriptional control is fundamental to cellular function. However, despite knowing that transcription factors can repress or activate specific genes, how these functions are implemented at the molecular level has remained elusive, particularly in the endogenous context of developing animals. Here, we combine optogenetics, single-cell live-imaging, and mathematical modeling to study how a zinc-finger repressor, Knirps, induces switch-like transitions into long-lived quiescent states. Using optogenetics, we demonstrate that repression is rapidly reversible (~1 min) and memoryless. Furthermore, we show that the repressor acts by decreasing the frequency of transcriptional bursts in a manner consistent with an equilibrium binding model. Our results provide a quantitative framework for dissecting the in vivo biochemistry of eukaryotic transcriptional regulation.

Throughout biology, transcription factors dictate gene expression and, ultimately, drive cell-fate decisions that play fundamental roles in development[1], immune responses[2], and disease[3]. Achieving a quantitative and predictive understanding of how this process unfolds over time and space holds the potential both to shed light on the molecular mechanisms that drive cellular decision-making and to lay the foundation for a broad array of bioengineering applications, including the synthetic manipulation of developmental processes[4–8] and the development of therapeutics[9].

In recent years, great progress has been made in uncovering the molecular mechanism of transcription factor action through cell culture-based methods thanks to the emergence of a wide array of imaging techniques that can query the inner workings of cells in real time, often at the single molecule level (see, for example, refs. 10–18). Building on these works, we and others have developed technologies that allow for the direct measurement of protein concentrations[19] and transcriptional dynamics[20–23] in single cells of living multicellular organisms, making it possible to study how transcription factors function in their endogenous context.

However, inferring regulatory mechanisms requires that these quantitative readouts be paired with time-resolved perturbations that push systems away from their wild-type trajectory. Optogenetic tools can address this need by enabling the manipulation of control of transcription factor function in vivo via the light-based modulation of nuclear protein concentration[24–34]. Yet, many existing optogenetics approaches either do not permit the direct control of transcription factor concentrations within nuclei or act on timescales of hours or days, limiting their utility for testing molecular models of gene regulatory function[35].

Here, we combine in vivo measurements of protein concentrations[19] and transcriptional dynamics[20] with an optogenetic system that permits sub-minute manipulation of nuclear protein concentrations[24,25,32–34]. We leverage our ability to rapidly measure and manipulate transcriptional systems to study causal connections between the molecular players that underpin transcriptional control,

[1]Department of Physics, University of California, Berkeley, CA, USA. [2]Biophysics Graduate Group, University of California, Berkeley, CA, USA. [3]Department of Plant and Microbial Biology, University of California, Berkeley, CA, USA. [4]Department of Molecular and Cell Biology, University of California, Berkeley, CA, USA. [5]Institute for Quantitative Biosciences-QB3, University of California, Berkeley, CA, USA. [6]Chan Zuckerberg Biohub, San Francisco, CA, USA. [7]Present address: Department of Genetics, Harvard Medical School, Boston, MA, USA. [8]Present address: Department of Pathology, Brigham and Women's Hospital, Boston, MA, USA. [9]Present address: Department of Genome Sciences, University of Washington, Seattle, WA, USA. [10]Present address: Feedstocks Division, Joint BioEnergy Institute, Emeryville, CA, USA. [11]Present address: Environmental Genomics and Systems Biology Division, LBNL, Berkeley, CA, USA. [12]These authors contributed equally: Jiaxi Zhao, Nicholas C. Lammers. ✉e-mail: hggarcia@berkeley.edu

shedding light on the molecular basis of transcriptional repression in a developing animal.

We use this platform to answer two key questions regarding the kinetic properties of repression. First, despite several studies dissecting repressor action at the bulk level[36–40], it is not clear whether this repression is implemented in a graded or switch-like fashion at individual gene loci over time (Fig. 1A, left). Second, the adoption of cellular fates—often dictated by repressors—has been attributed to the irreversible establishment of transcriptional states[2,41,42]. However, minute-resolution measurements tracking the timescales over which reversible repressor binding induces long-lived, irreversible transcriptional inactivity have been lacking. Is the action of repressors itself reversible over relevant developmental timescales—such that sustained repressor binding is required to maintain gene inactivity—or does repression almost immediately become irreversible—such that even transient exposure to high repressor concentrations is sufficient to induce long-lived transcriptional inactivity (Fig. 1A, right)?

In this work, we put these questions to the quantitative test, examining how the zinc-finger repressor Knirps drives the formation of stripes 4 and 6 of the widely studied *even-skipped* (*eve*) pattern during

the early development of the fruit fly *Drosophila melanogaster* (Fig. 1B)[43–45]. By integrating our optogenetic platform with quantitative modeling, we have elucidated previously unexplored aspects of the molecular basis of in vivo transcriptional control. Our findings reveal that Knirps repression operates in a switch-like manner, is rapidly reversible, and lacks transcriptional memory. Furthermore, we demonstrate that this repression mechanism is mediated by a reduction in the frequency of transcriptional bursts.

## Results

### An optogenetics platform for dissecting single-cell repression dynamics in development

To measure Knirps protein concentration dynamics, we labeled the endogenous *knirps* locus with a LlamaTag, a fluorescent probe capable of reporting on protein concentration dynamics faster than the maturation time of more common fluorescent protein fusions[19]. Further, we quantified the target transcriptional response using a reporter construct of the *eve* stripe 4 + 6 enhancer[43], where the nascent RNA molecules are fluorescently labeled using the MCP-MS2 system[20,21,46] (Fig. 1C). The resulting nuclear fluorescence and transcriptional puncta

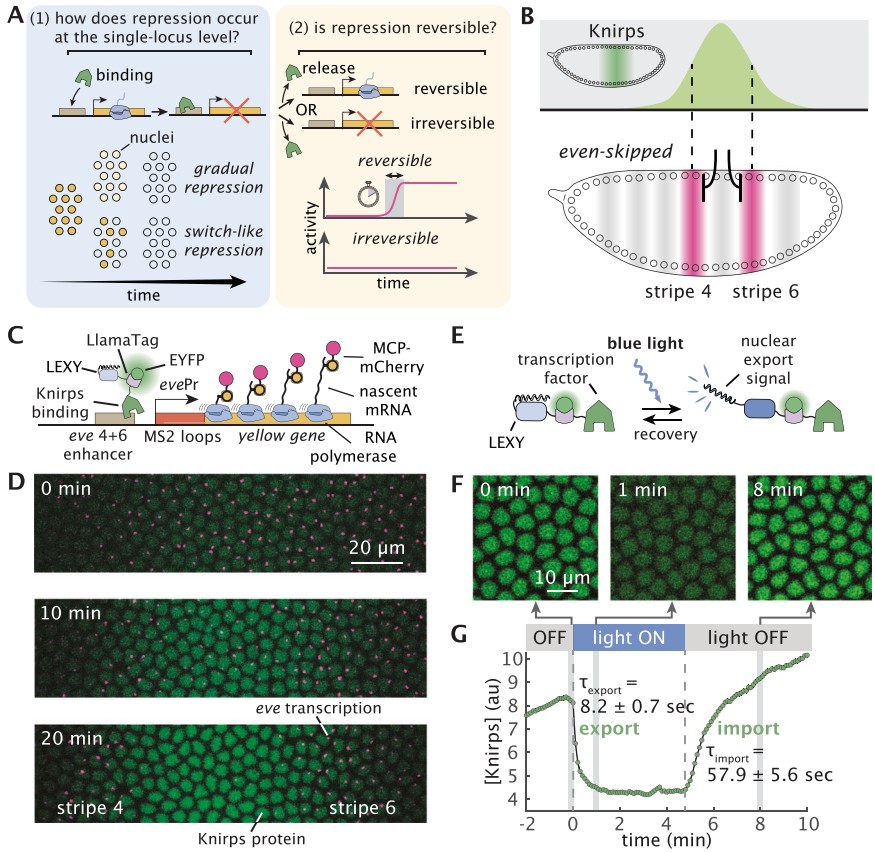

**Fig. 1 | Combining optogenetics and live imaging enables dissection of single-cell repression dynamics in a developing animal. A** Key questions regarding transcriptional repression. Left: Whether single-cell repression occurs in a gradual or switch-like fashion over time. Right: Whether repression is reversible. **B** Knirps represses *even-skipped* (*eve*) stripes 4 + 6 transcription in the fruit fly embryo. Top: Knirps is expressed in a bell-shaped domain during early embryogenesis. Bottom: Knirps specifies the position and sharpness of the inner boundaries of *eve* stripes 4 and 6. **C** Two-color tagging permits the simultaneous visualization of input protein concentration and output transcriptional dynamics in vivo. Maternally deposited EYFP molecules bind to Knirps-LlamaTag, resulting in increased nuclear fluorescence, which provides a real-time readout of the nuclear protein concentration. Maternally deposited MS2 coat protein (MCP) binds to MS2 stem-loops in the nascent RNA formed by RNAP molecules elongating along the body of the *eve* 4 + 6 reporter construct leading to the accumulation of fluorescence at sites of nascent

transcript formation. LEXY tag is also fused to Knirps to allow for optogenetic manipulation of its nuclear concentration. **D** Representative frames from live-imaging data. The embryo is oriented with the anterior (head) to the left. Green and magenta channels correspond to Knirps repressor and *eve* 4 + 6 transcription, respectively. When Knirps concentration is low, *eve* stripe 4 + 6 is expressed in a broad domain, which refines into two flanking stripes as Knirps concentration increases. **E** Optogenetic control of nuclear protein export. Upon exposure to blue light, the nuclear export signal within the LEXY domain is revealed. As a result, the fusion protein is exported from the nucleus. **F** Fluorescence images of embryos expressing the Knirps-LEXY fusion undergoing an export-recovery cycle. **G** Relative nuclear fluorescence of the repressor protein over time ($n$ = 55 nuclei). Half-times for export and recovery processes are estimated by fitting the fluorescence traces to exponential functions.

provide a direct readout of input Knirps concentration and output *eve* 4 + 6 transcription, respectively, as a function of space and time (Fig. 1D; Supplementary Movie 1). Our data recapitulate classic results from fixed embryos[47] in dynamical detail: gene expression begins in a domain that spans stripes 4 through 6, subsequently refined by the appearance of the Knirps repressor in the interstripe region.

To enable the precise temporal control of Knirps concentration, we attached the optogenetic LEXY domain[24] to the endogenous *knirps* locus in addition to the LlamaTag (Fig. 1C). Upon exposure to blue light, the LEXY domain undergoes a conformational change which results in the rapid export of Knirps protein from the nucleus (Fig. 1E). Export-recovery experiments revealed that export dynamics are fast, with a half-time < 10 s, while import dynamics are somewhat slower, with a half-time ~ 60 s upon removal of illumination (Fig. 1F, G; Supplementary Movie 2). These time scales are much faster than typical developmental time scales[48], allowing us to disentangle rapid effects due to direct regulatory interactions between Knirps and *eve* 4 + 6 from slower, indirect regulation that is mediated by other genes in the regulatory network. We established stable breeding lines of homozygous optogenetic Knirps flies, demonstrating that the protein tagged with both LEXY and LlamaTag is homozygous viable. Furthermore, our optogenetic Knirps drives comparable levels of *eve* 4 + 6 than wild-type Knirps (Supplementary Fig. 1). Thus, we conclude that our optogenetics-based approach represents an ideal platform for manipulating transcriptional systems to probe the molecular basis of gene regulatory control without significantly affecting the broader

regulatory network and the developmental outcome this network encodes for.

## Repressor concentration dictates transcriptional activity through all-or-none response

To understand how Knirps repressor regulates *eve* 4 + 6 expression, we first analyzed the temporal dynamics of Knirps-LlamaTag-LEXY (hereafter referred to simply as "Knirps") concentration and *eve* 4 + 6 expression in the absence of optogenetic perturbations. We generated spatiotemporal maps of input repressor concentration and output transcription by spatially aligning individual embryos according to the peak of the Knirps expression domain along the anterior-posterior axis (Supplementary Figs. 2, 3 and 4). These maps reveal a clear pattern: rising repressor concentrations coincide with a sharp decline in *eve* 4 + 6 activity at the center of the Knirps domain. To further investigate the regulatory impact of Knirps, while minimizing the influence of other regulatory factors, we focused in on this central region of the Knirps domain (-2% to 2% of the embryo length with respect to the center of the domain). Here, we observe a clear anti-correlation between Knirps concentration, which increases steadily with time, and the mean transcription rate, which drops precipitously between 10 and 20 min into nuclear cycle 14 (Fig. 2A).

We quantified the regulatory relationship implied by these trends by calculating the Knirps vs. *eve* 4 + 6 "input-output function", which reports on the average transcription rate as a function of nuclear repressor concentration (inset panel in Fig. 2A; Supplementary Fig. 5).

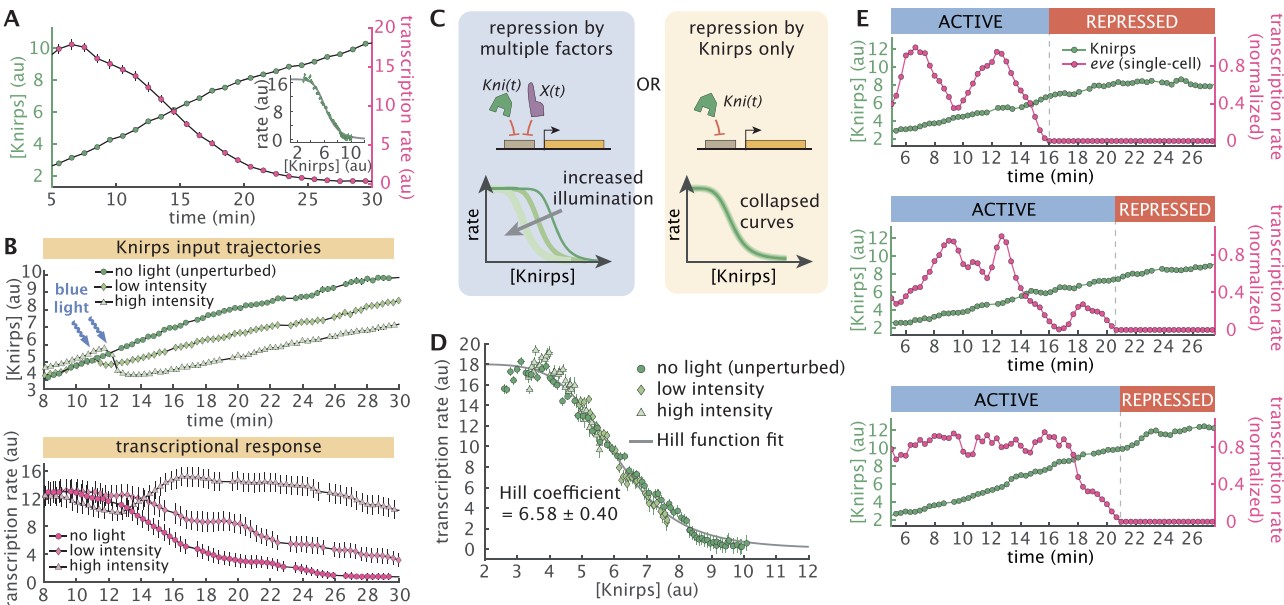

**Fig. 2 | Knirps concentration dictates sharp, switch-like repression. A** Average Knirps concentration (green) and *eve* 4 + 6 transcription (magenta) over time shows a clear anticorrelation. These dynamics are calculated by averaging the traces over a window of -2% to 2% along the anterior-posterior axis of the embryo and centered around the peak of the Knirps pattern (Supplementary Fig. 2). Target transcription declines sharply as Knirps concentration increases. Inset panel shows the input-output relationship under this no light (unperturbed) condition. **B** Optogenetics allows for titration of protein concentration. Top panel shows the average Knirps concentration for three embryos, each under different illumination intensities. Bottom panel shows the corresponding trends in the *eve* 4 + 6 transcription rate. The illumination started around 12 min into nuclear cycle 14 and continued throughout the experiment. **C** To test whether Knirps is the only repressor whose concentration changes in the system, input-output functions under different illumination conditions can be compared. If there are multiple potentially unknown repressors at play (e.g. the *X* transcription factor in the figure), then each illumination level should lead to a different input-output function (left). However, if

Knirps is the sole repressor, the input-output functions for each condition should collapse onto a single curve (right). **D** Average transcription rate as a function of Knirps concentration for each illumination condition (averaged over a window of -2% to 2% along the anterior-posterior axis). All three conditions follow the same trend, suggesting that Knirps is the only repressor regulating target transcription during this developmental stage. The input-output relationship is fitted with a Hill function resulting in a Hill coefficient of 6.58. (Averaged over $n = 4$ for no light, $n = 4$ for low intensity and $n = 3$ for high intensity embryos.) **E** Illustrative single-cell Knirps (green points) and transcriptional dynamics (magenta points) show that repression is switch-like at the single-cell level. Traces are normalized by their maximum transcription rate and smoothened using a moving average of 1 min. (Error bars in **A**, **B**, and **D** indicate the bootstrap estimate of the standard error. $t = 0$ is defined as the onset of transcription in nuclear cycle 14. Transcription rate reflects the measured MS2 signal, which is an approximation of the *eve* mRNA production rate[19,20,57].)

This measurement revealed a sharp decline in transcriptional activity across a narrow band of Knirps concentrations, suggesting that *eve* 4 + 6 loci are highly sensitive to nuclear repressor levels. This finding is consistent with previous observations that Knirps represses *eve* 4 + 6[49], and with the discovery of multiple Knirps binding sites in the *eve* 4 + 6 enhancer region (Supplementary Fig. 6)[50]. However, neither our endogenous measurements nor these previous studies can rule out the possibility that other repressors might also play a role in driving the progressive repression of *eve* 4 + 6 over the course of nuclear cycle 14. Indeed, by themselves, neither live imaging experiments (which are constrained to observing wild-type trends) nor classical mutation-based studies (which are subject to feedback encoded by the underlying gene regulatory network) can rule out the presence of other inputs.

Our optogenetics approach allows us to circumvent these limitations and search for regulatory inputs that impact *eve* 4 + 6 expression, but are not directly observed in our experiments. Specifically, we used optogenetics to alter Knirps concentration dynamics over the course of nuclear cycle 14. Shortly after the beginning of the nuclear cycle, we exposed embryos to low and high blue light illumination, inducing moderate and strong reductions in nuclear Knirps concentration, respectively, which resulted in distinct transcriptional trends (Fig. 2B; Supplementary Fig. 7; Supplementary Movie 3). We reasoned that, because we are only altering Knirps concentration dynamics, the presence of other repressors dictating *eve* 4 + 6 activity together with Knirps should lead to distinct input-output curves across these different illumination conditions (Fig. 2C, left). Conversely, if Knirps is the sole repressor driving the repression of *eve* 4 + 6 over time, the transcriptional input-output function should be invariant to perturbations of Knirps concentration dynamics (Fig. 2C, right).

Comparing the *eve* 4 + 6 vs. Knirps input-output function for the unperturbed control (inset panel of Fig. 2A) to that of optogenetically perturbed embryos (Fig. 2D), we find that all three conditions collapse onto a single input-output curve, providing strong evidence that Knirps is the sole repressor of *eve* 4 + 6. Moreover, as noted above, we find that Knirps repression occurs in a sharp fashion: *eve* 4 + 6 loci transition from being mostly active to mostly repressed within a narrow band of Knirps concentrations. To quantify this sharp response, we fit a Hill function to the data in Fig. 2D (gray line), which yielded a Hill coefficient of 6.58 ± 0.40. Notably, this is comparable to Hill coefficients estimated for the Bicoid-dependent activation of *hunchback*[51–53]; another canonical example of sharp gene regulation—in this case, of activation—during developmental patterning which relies on the presence of multiple binding sites for the transcription factor within the enhancer.

The input-output function in Fig. 2D summarizes the average effect of repressor level on *eve* 4 + 6 expression, but it cannot alone shed light on how this effect is achieved in individual cells. Thus, we next investigated how this sharp average decrease in gene expression is realized at the single-cell level. We examined single-cell trajectories of Knirps repressor and corresponding *eve* 4 + 6 transcription. This revealed that the sharp population-level input-output function illustrated in Fig. 2D is realized in an all-or-none fashion at the level of individual cells (Fig. 2E; Supplementary Fig. 8). During this process, the gradual rise in Knirps concentration induces an abrupt, seemingly irreversible, transition from active transcription to a long-lived (or even permanent), transcriptionally quiescent state.

### Rapid export of repressor reveals fast, reversible reactivation kinetics at the single-cell level

It has been shown that the activity of repressors can have different degrees of reversibility[13,54]. For example, recruitment of certain chromatin modifiers may silence the locus even if the initial transcription factor is no longer present[13]. The single-cell traces in Fig. 2E and Supplementary Fig. 8 appear to transition into an irreversible transcriptional quiescent state. However, since Knirps concentration keeps increasing after *eve* 4 + 6 expression shuts off, it is possible that repression is, in fact, reversible and that the observed irreversibility is due only to the monotonic increase of the repressor concentration over time.

To probe the reversibility of Knirps-based repression, we used optogenetics to induce rapid, step-like decreases in nuclear Knirps concentration (Fig. 3A). Prior to the perturbation, the system was allowed to proceed along its wild-type trajectory until the majority of *eve* 4 + 6 loci at the center of the Knirps domain were fully repressed. Strikingly, when blue light was applied to export Knirps, we observed a widespread, rapid reactivation of repressed *eve* loci (Fig. 3B and C; Supplementary Movie 4). To probe the time scale of reactivation, we calculated the fraction of active nuclei as a function of time since Knirps export (Fig. 3D, Supplementary Figs. 9 and 10). This revealed that *eve* loci begin to reactivate in as little as 1 min following illumination. We obtain a reactivation time distribution from single-cell trajectories with a mean response time of 2.5 min (Fig. 3E) and find that transcription fully recovers within 4 min of Knirps export (Fig. 3D). Thus, Knirps repression is completely reversible.

Previous studies have revealed regulatory "memory" wherein the repressive effect of certain repressors increases with longer exposure[13]. Thus, we reasoned that prolonged exposure to high levels of a repressor could induce the accumulation of specific chemical or molecular modifications that prevent activator binding and, as a result, impede reactivation at the target locus, such as histone modifications[55]. If this process is present, we should expect gene loci that have been repressed for a longer period before optogenetically triggering repressor export to require more time to reactivate. To test this hypothesis, we used the measured single-cell reactivation trajectories (Fig. 3C) to calculate the average reactivation time as a function of how long cells had been repressed prior to Knirps export. Interestingly, our analysis reveals that the reactivation time has no dependence on the repressed duration (Fig. 3F). This, combined with the fact that nearly all (97%) repressed gene loci reactivate upon Knirps export (inset panel in Fig. 3E), argues against the accumulation of any significant molecular memory amongst repressed gene loci within the ~10 min time scale captured by our experiments. Instead, it points to a model where repressor action is quickly reversible and memoryless.

### Knirps acts by inhibiting the initiation of transcription bursts

One of the simplest models that can capture the reversible, memoryless transitions between active and inactive transcriptional states observed in Fig. 3 is a two-state model, in which the gene promoter switches stochastically between periods of transcriptional activity ("bursts") and periods of inactivity[45,52,56–61]. Here, the gene promoter switches between active (ON) and inactive (OFF) states with rates $k_{on}$ and $k_{off}$, and initiates RNAP molecules at a rate $r$ while in the ON state (Fig. 4A). Consistent with this model, our single-cell transcriptional traces show clear signatures of transcriptional bursting (see, e.g., top two panels of Fig. 2E; Supplementary Fig. 8), suggesting that this two-state framework provides a viable basis for examining how Knirps regulates transcriptional activity at *eve* 4 + 6 loci.

Within this model, the repressor can act by decreasing burst frequency (decreasing $k_{on}$), by decreasing the duration of transcriptional bursts (increasing $k_{off}$), by decreasing the burst amplitude (decreasing $r$), or any combination thereof as shown in Fig. 4A. To shed light on the molecular strategy by which Knirps represses *eve* 4 + 6, we utilized a recently-developed computational method that utilizes compound-state Hidden Markov Models (cpHMM) to infer promoter state dynamics and burst parameter values ($k_{on}$, $k_{off}$, and $r$) from single-cell transcriptional traces as a function of Knirps concentration (Fig. 4B)[57]. We used data from all three illumination conditions (outlined in Fig. 2B) and conducted burst parameter inference on 15 min-long segments of MS2 traces sampled from nuclei falling within the center

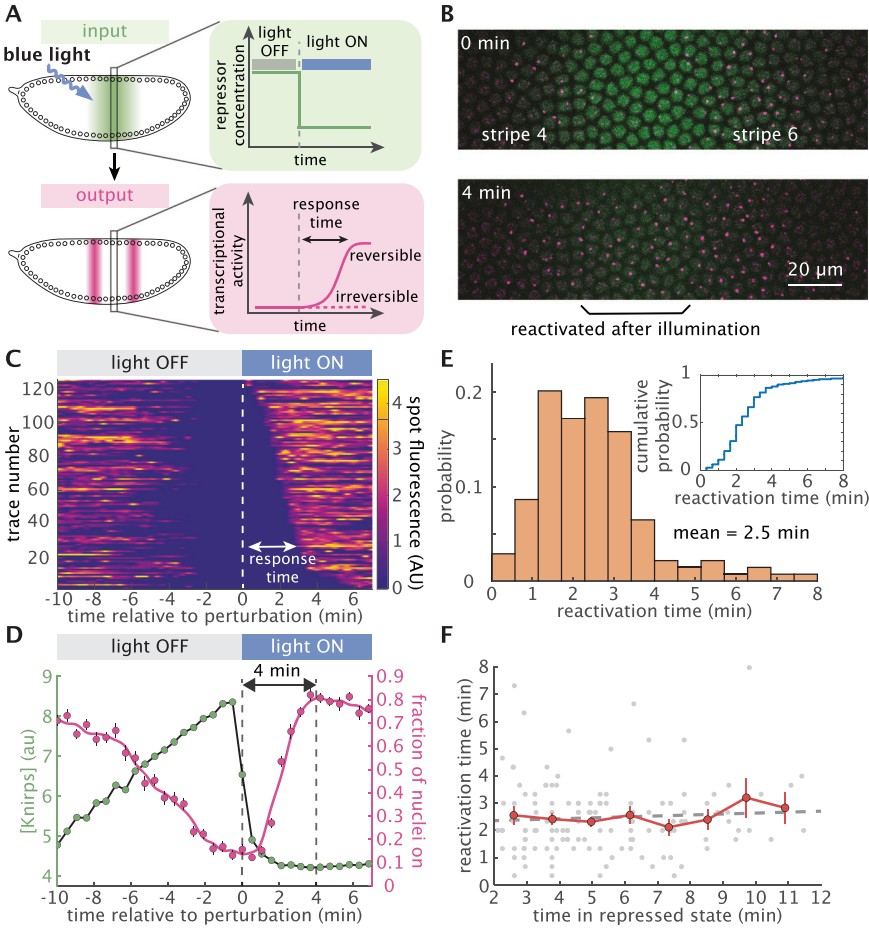

**Fig. 3 | Knirps repression is rapidly reversible and memoryless. A** Testing the reversibility of Knirps repression using a step-like optogenetic perturbation. Upon removal of Knirps repressor from the nucleus, transcriptional activity can remain repressed or recover, depending on whether repression is irreversible or reversible. **B** Snapshots from a movie before (top) and after (bottom) the optogenetic export of Knirps protein. Nuclei whose transcription was originally repressed by Knirps fully reactivate after 4 min of illumination. **C** Heatmap of single-cell reactivation trajectories sorted by response time. Response time is defined as the interval between the perturbation time and when the MS2 spots reappear. **D** Average repressor concentration (green) and the fraction of actively transcribing cells (magenta) before and after blue light illumination. We find that Knirps repression is rapidly reversible within 4 min. ($n = 229$ nuclei from 4 embryos, averaged over a -2% to 2% window along the anterior-posterior axis centered on the Knirps concentration peak). **E** Fast reactivation occurs with an average of 2.5 min. The

reactivation response time is calculated as the interval between the perturbation and when a locus is first observed to resume transcription. ($n = 139$ nuclei from 4 embryos). Inset panel describes the cumulative distribution of reactivation times. To exclude gene loci that were transiently OFF due to transcriptional bursting or missed detections, we focused this analysis on gene loci that were silent for at least 2 min before perturbation. **F** Knirps repression is memoryless. Plot showing the reactivation response time of individual loci as a function of the time spent in the repressed state before optogenetic reactivation. Fitting with a linear regression model (gray dotted line) results in $p$-value = 0.495, which confirms that the reactivation response time is independent of the repressed duration of the locus. Red dots represent the means of the binned data. (Error bars in **D** and **F** indicate the bootstrap estimate of the standard error. $p$-value in F is for the $F$-test on the regression model, which tests whether the model fits significantly better than a degenerate model consisting of only a constant term).

of the Knirps domain (-2% to 2% of the embryo length with respect to the center of the domain).

To reveal burst parameter dependence on Knirps concentration, we grouped traces based on low ([Knirps] $\leq 4$ au) and high ([Knirps] $\geq 6$ au) Knirps concentrations (Fig. 4B) and conducted cpHMM inference. We find that the repressor strongly impedes locus activation, decreasing the frequency of transcriptional bursts ($k_{on}$) from 2.3 bursts per minute down to 1.1 burst per minute between low and high Knirps concentrations (Fig. 4C, left panel). We also find a moderate (~30%) increase in the duration of transcriptional bursts between low and high Knirps concentrations; however this change is smaller than the uncertainty in our inference (Fig. 4C, middle panel). Finally, we find no significant change in the burst amplitude as a function of Knirps concentration (Fig. 4C, right panel). Thus, burst parameter inference indicates that Knirps represses *eve* 4 + 6 loci mainly by interfering with the initiation of transcriptional bursts. See

Supplementary Note 1 and Supplementary Fig. 11 for additional cpHMM inference results.

To our knowledge, Fig. 4C provides the unprecedented simultaneous measurement of transcription factor concentration and burst dynamics in a living multicellular organism. However, these results are, necessarily, a coarse-grained approximation of the true regulatory dynamics. This is because our cpHMM inference has an inherently low temporal resolution, reflecting averages taken across 15 min periods of time and across large ranges of input Knirps concentrations. However, in principle, our live imaging data—which contains high-resolution time traces of both input repressor concentration dynamics and output transcriptions rates—should make it possible to move beyond these coarse-grained estimates to recover the true, instantaneous regulatory relationship between Knirps concentration and burst dynamics.

To answer these questions, we developed a comprehensive computational method that utilizes stochastic simulations of single-

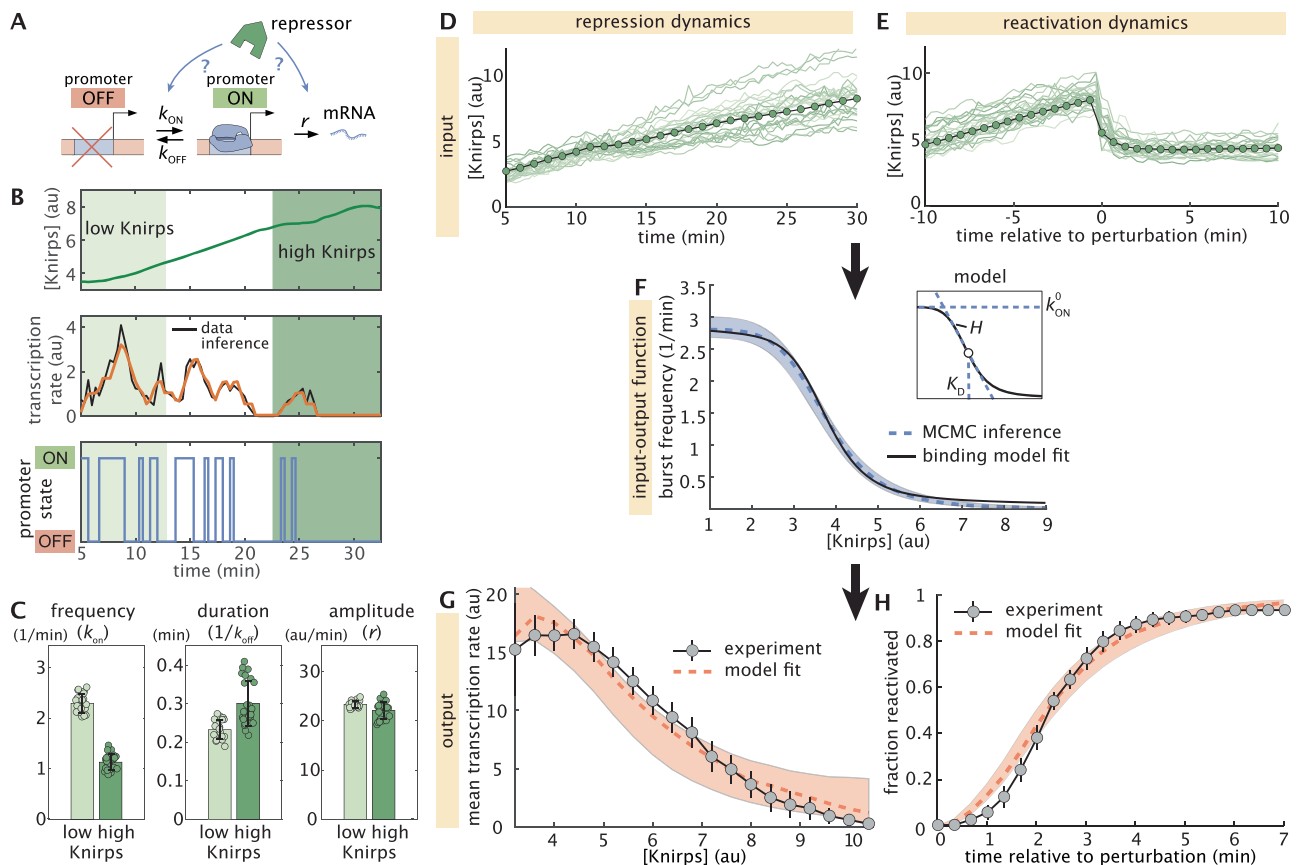

**Fig. 4 | Knirps represses through rapid modulation of burst frequency. A** Cartoon illustrating the two-state bursting model where a promoter can stochastically transition between active and inactive states. **B** A representative experimental trace of Knirps protein (top) and transcription dynamics, along with the best fit (middle) and the corresponding sequence of inferred promoter activity states (bottom) returned by cpHMM inference. **C** Bar plots indicating cpHMM burst parameter inference results for *eve* 4 + 6 loci subjected to low (≤4 au) and high (≥6 au) Knirps concentrations. Circles indicate individual bootstrap inference replicates.

**D–H** Summary of stochastic simulation methodology and results. **D** Illustrative individual (green lines) and average (green circles) nuclear Knirps concentration trajectories as a function of time in unperturbed embryos. **E** Individual and average nuclear Knirps concentrations before and after optogenetic export, which happens at time $t = 0$. **F** We take $k_{on}$ to be a Hill function of Knirps concentration, with a shape that is determined by three microscopic parameters: $k_{on}^0$, $K_D$, and $H$ (see inset panel and Equation (1)). The dashed blue curve indicates the input-output function

for the burst frequency trend ($k_{on}$) corresponding to the best-fitting set of microscopic parameters. The black line shows the best-fitting curve predicted by an equilibrium binding model with 10 Knirps binding sites. **G** Modeling results for the average fluorescence and (**H**) reactivation dynamics as a function of Knirps concentration. Dashed red line indicates the prediction of the best-fitting model realization. (Error bars in (**C**) reflect the standard error of the mean, as estimated from 21 (low group) and 25 (high group) bootstrap burst inference replicates conducted on MS2 traces from 9 embryos. In (**G**) the transcription rate is calculated from the measured MS2 signal, which is an approximation of the mRNA production rate[19,20,57]. Error bars in G and H indicate the bootstrap standard error estimated using 100 bootstrap samples of MS2 traces from 11 and 4 embryos, respectively. Shaded regions in (**F**, **G**) and H indicate "1 sigma" uncertainty range as estimated from 54,000 MCMC samples of model parameters. Dashed orange indicate the $k_{on}$ curve (F), average fluorescence (G), and reactivation (H) trends for the 25 best-fitting model realizations).

cell transcriptional trajectories to test theoretical model predictions against our experimental measurements and uncover repressor-dependent burst parameter trends (Supplementary Fig. 12; Supplementary Note 2). Motivated by the cpHMM inference shown in Fig. 4C, as well as by finer-grained results shown in Supplementary Fig. 11, we allow both the burst frequency and the burst duration (but not the burst amplitude) to vary as a function of Knirps concentration. We assume a model in which these parameters are simple Hill functions of repressor concentration. For the burst frequency ($k_{on}$), this leads to a function with the form

$$k_{on}([\text{Knirps}]) = k_{on}^0 \frac{K_D^H}{[\text{Knirps}]^H + K_D^H}, \qquad (1)$$

where $k_{on}^0$ sets the maximum burst frequency value, the Hill coefficient $H$ sets the sharpness of the response, and $K_D$ dictates the Knirps concentration midpoint for the transcriptional response, giving the

repressor concentration where $k_{on}$ drops to half its maximum value. Together, these "microscopic" parameters define an input-output function that directly links the burst frequency to Knirps concentration. As noted above, we also allow the burst duration to vary as a function of Knirps concentration (see Supplementary Equation (2) and Supplementary Note 2.1 for further details). However we focus on $k_{on}$ throughout the main text, since it is the only parameter that decreases as a function of Knirps concentration (and, thus, the only parameter that could drive *eve* 4 + 6 repression).

With our model defined, our procedure is as follows: we start by sampling real single-cell Knirps concentration trajectories from (i) the three illumination conditions shown in Fig. 2D and (ii) the reactivation experiments shown in Fig. 3 (Fig. 4D and E, respectively). Then, we plug these Knirps trajectories into the input-output functions defined in Equation (1) (for burst frequency; see also Fig. 4F) and Supplementary Equation (2) (for burst duration). Next, given a set of microscopic parameters (e.g., $H$, $K_D$, and $k_{on}^0$ for Equation (1)), we generate time-

dependent burst parameter trends (Supplementary Fig. 12B). We then use these trends to simulate corresponding ensembles of MS2 traces (Supplementary Fig. 12C–F; see also Supplementary Note 2.1). We use these simulated MS2 traces to calculate, first, the predicted Knirps vs. *eve* 4 + 6 input-output function (Fig. 4G) and, second, the predicted reactivation cumulative distribution function curve (Fig. 4H). Finally, we compare these predictions to empirical measurements of the same quantities from our live imaging experiments (see Fig. 2D and inset panel of Fig. 3E). Through this process of simulation and comparison, each set of microscopic parameters used to calculate our predictions are assigned a fit score. We then use parameter sweeps and Markov Chain Monte Carlo (MCMC)[62,63] to search for parameters that most successfully reproduce our live imaging results (see Supplementary Fig. 12E–G and Supplementary Notes 2.3 and 2.4).

As illustrated in Fig. 4F, we find that the best-fitting model features a sharp $k_{on}$ versus Knirps input-output function ($H$ = 6.1 ± 0.7). We also find that $k_{on}$ has a relatively low $K_D$ of 3.7 au ± 0.1 with respect to the range of Knirps concentrations experienced by *eve* 4 + 6 loci (see Fig. 2B, bottom), which implies that gene loci have a low concentration threshold for Knirps repression. As a result of this low threshold, *eve* 4 + 6 loci are effectively clamped in the OFF state ($k_{on} \leq 0.1$ bursts per minute) once the Knirps concentration exceeds 6 au, which happens about 12 min into nuclear cycle 14 for the average nucleus at the center of the Knirps domain (Fig. 2B, bottom). Finally, while burst duration does not play a role in *eve* 4 + 6 repression, our results indicate that a moderate Knirps-dependent increase in burst duration is required in order to explain our experimental data (Supplementary Fig. 13). See Supplementary Fig. 14 and Supplementary Note 2.5 for full inference results. Our findings also demonstrate that a simple two-state model in which Knirps represses *eve* 4 + 6 by decreasing the frequency of transcriptional bursts is sufficient to quantitatively recapitulate both the sharp decrease in the average transcription rate with increasing Knirps concentration (Fig. 4G) and the kinetics of reactivation following Knirps export (Fig. 4H).

Our simulation results also shed further light on the dynamics of *eve* reactivation following the step-like optogenetic export of Knirps protein from the nucleus (Fig. 3A). From Fig. 3E and F, we know that it takes ~2–4 min following Knirps export for MS2 spots to reappear in our live-imaging experiments. Yet this is the time scale for detection—for the amount of time it takes for genes to produce detectable levels of transcription and, hence, MS2 fluorescence—and thus likely overestimates the true *eve* 4 + 6 response time. So how fast is it really? Our model, which accounts for the fluorescence detection limit, predicts that $k_{on}$ recovers to half of its steady-state value within 30 s of the start of the optogenetic perturbation (Supplementary Fig. 15). Furthermore, we predict that half of all gene loci switch back into the transcriptionally active (ON) state within 102 s (1.7 min). Thus, it takes fewer than 2 min for *eve* 4 + 6 loci to "escape" Knirps repression and re-engage in bursty transcription.

## Discussion

Taken together, our results point to a model wherein the repressor acts upon the gene locus while it is transcriptionally inactive (OFF) to inhibit re-entry into the active (ON) state. Consistent with this picture, we find that the functional relation between $k_{on}$ and Knirps concentration inferred by MCMC inference is well explained by a simple equilibrium binding model where the burst frequency is proportional to the number of repressor molecules bound at the 4 + 6 enhancer (solid black curve in Fig. 4F; see Supplementary Note 3 for details).

Our in vivo dissection provides important clues toward unraveling the molecular basis of repressor action. We show that Knirps repression is switch-like (Fig. 2), memoryless (Fig. 3F), and rapidly reversible (Fig. 3E). Another key point is that, although our model predicts that gene loci require 1–2 min to reactivate and enter the ON state following the optogenetic export of Knirps from the nucleus (Supplementary

Fig. 15), the model assumes that the burst frequency itself responds instantaneously to changing Knirps concentration (see Equation (1), blue curve in Supplementary Fig. 15). While no reaction can truly be instantaneous, the success of this model in describing repression dynamics points to an underlying mechanism controlling the burst frequency that rapidly reads and responds to changing repressor concentrations, likely within a matter of seconds—a timescale that is consistent with the fast binding and unbinding dynamics reported for eukaryotic transcription factors[64].

We also note that the success of the two-state bursting model (Fig. 4A) at recapitulating Knirps repression dynamics (Fig. 4G and H) suggests that the same molecular process may be responsible for both the short-lived OFF periods between successive transcriptional bursts (see, e.g., the middle panel of Fig. 4B) and the much longer-lived periods of quiescence observed in repressed nuclei (e.g., Fig. 3C), and that there may be no need to invoke an "extra" repressor-induced molecular state outside of the bursting cycle[65–67]. At the same time, we cannot rule out the presence of additional, rapid kinetic steps both in the transcriptional bursting cycle and in the reactivation pathway. For reactivation times in particular, several factors, such as the rapid dynamics of repressor concentrations and our limited sensitivity for the detection of dim transcriptional spots, add complexity to the task of identifying the correct theoretical model for describing the experimental data. We anticipate that future refinements to experimental and theoretical approaches put forward in this work will be critical to further elucidating the kinetics of transcriptional regulation.

Previous work has established that Knirps plays a role in recruiting histone deacetylase[68] and that Knirps repression coincides with increased histone density at target enhancers such as the one dissected here[38]. This suggests a model in which the repressor modulates the longevity of the OFF state by tuning the accessibility of enhancer DNA, which would impact activator binding, and also indicates that Knirps cannot act to repress the locus during active bursts. It is notable, however, that the 1–2 min reactivation time scales revealed (Fig. 3; Supplementary Fig. 15) are faster than most chromatin-based mechanisms measured in vivo so far[13,54,64,69,70]. This rapid reversibility, along with the memoryless nature of Knirps repression, indicates that whatever the underlying mechanism, Knirps binding at the locus is necessary in order to maintain the gene in a transcriptionally inactive state at the stage of development captured by our live imaging experiments. Interestingly, we found that the modulation of burst frequency by Knirps can be recapitulated by a simple thermodynamic model predicting Knirps DNA occupancy (black line in Fig. 4F; see Supplementary Note 3 for further details). This suggests that the wide repertoire of theoretical and experimental approaches developed to test these models (see, for example,[71]) can be used to engage in a dialog between theory and experiment aimed at dissecting the molecular mechanism underlying the control of transcriptional bursting.

Critically, none of these molecular insights would have been possible without the ability to measure and acutely manipulate input transcription factor concentrations in living cells. Thus, by building on previous works using the LEXY technology in different biological contexts[24,25,32–34], our work demonstrates the power of the LEXY system for simultaneously manipulating—and measuring—nuclear protein concentrations and the resulting output transcriptional activity. Supplementary Note 4 outlines how the LEXY system improves upon many previously reported methods of optogenetic control in embryos[26–31,72–75].

More work remains, however. Optogenetic tools with better dynamical range could open the door to studies of enhancers that respond sensitively to low transcription factor concentrations, providing an even more powerful approach for probing gene regulatory logic. Additionally, our optogenetic system's capacity for time-resolved measurements of transcriptional inputs and outputs in single cells raises the possibility of investigating the noise characteristics

of gene-regulatory systems. As an example, Supplementary Note 5 demonstrates how the two state bursting model can be used to make simple theoretical predictions about the noise levels for different repression strategies. Future work will seek to refine the experimental and theoretical tools presented here with the aim of reliably measuring and interpreting transcriptional heterogeneity in vivo.

Looking ahead, we anticipate that our live imaging approach, along with the quantitative analysis framework presented in this work, will provide a useful foundation for similar in vivo biochemical dissections of how the transcription factor-mediated control of gene expression dictates transcriptional outcomes, opening the door to a number of exciting new questions relating to transcriptional regulation, cell-fate decisions, and embryonic development that span multiple scales of space and time.

## Methods

### Cloning and Transgenesis
The fly lines used in this study were generated by inserting transgenic reporters into the fly genome or by CRISPR-Cas9 genome editing, as described below. See Supplementary Table 1 for detailed information on the plasmid sequences used in this study.

**Creation of tagged _knirps_ loci using CRISPR-Cas9.** To tag endogenous the _knirps_ locus with the EGFP-LlamaTag and LEXY modules, we used CRIPSR-mediated homology-directed repair with donor plasmids synthesized by Genscript. gRNA was designed using target finder tool from flyCRISPR (https://flycrispr.org), and cloned based on the protocol from[76]. A yw;nos-Cas9(II-attP40) transgenic line was used as the genomic source for Cas9, and the embryos were injected and screened by BestGene Inc.

**Creation of _eve_ 4+6 reporter.** The _eve_ 4 + 6 enhancer sequence is based on 800 bp DNA segment described in[49]. The _eve_ 4 + 6 reporter was constructed by combining the enhancer sequence with an array of 24 MS2 stem-loops fused to the _D. melanogaster yellow_ gene[19]. The _eve_4+6-MS2-_Yellow_ construct was synthesized by Genscript and injected by BestGene Inc into _D. melanogaster_ embryos with a _ΦC31_ insertion site in chromosome 2L (Bloomington stock #9723; landing site VK00002; cytological location 28E7).

**Transgenes expressing EYFP and MCP-mCherry.** The fly line maternally expressing MCP-mCherry that is attached to a nuclear localization signal (chromosome 3) was constructed as described in[19]. The fly line maternally expressing EYFP (chromosome 2) was constructed as previously described in[77]. To simultaneously image protein dynamics using LlamaTags and transcription using MCP-MS2 system, we combined the vasa-EYFP transgene with MCP-mCherry to construct a new line (yw; vasa-EYFP; MCP-mCherry) that maternally expresses both proteins.

### Fly lines
To measure the Knirps pattern and corresponding _eve_ 4 + 6 transcription simultaneously, we performed crosses to generate female virgin flies carrying transgenes that drive maternal EYFP, MCP-mCherry, along with LlamaTag-LEXY tagged Knirps locus (yw; vasa-EYFP; MCP-mCherry/Knirps-LlamaTag-LEXY). These flies were then crossed with males having both the _eve_ 4 + 6 reporter and LlamaTag-LEXY tagged Knirps locus (yw; eve4+6-MS2-Yellow; Knirps-LlamaTag-LEXY). This resulted in embryos homozygous or heterozygous for the tagged Knirps locus also carrying maternally deposited EYFP, MCP-mCherry, and a _eve_ 4 + 6 reporter. Embryos homozygous for tagged Knirps can be differentiated from heterozygous embryos through a comparison of their nuclear fluorescence levels as shown in Supplementary Fig. 16. All the fly lines used in this work can be found in Supplementary Table 2.

### Embryo preparation and data collection
The embryos were prepared following procedures described in[19,20,57]. Embryos were collected and mounted in halocarbon oil 27 between a semipermeable membrane (Lumox film, Starstedt, Germany) and a coverslip. Confocal imaging on a Zeiss LSM 780 microscope was performed using a Plan-Apochromat 40x/1.4NA oil immersion objective. EYFP and MCP-mCherry were excited with laser wavelengths of 514 nm (3.05 $\mu$W laser power) and 594 nm (18.3 $\mu$W laser power), respectively. Modulation of Knirps nuclear concentration was performed by utilizing an additional laser with a wavelength of 458 nm, with laser power of 0.2 $\mu$W (low intensity in Fig. 2) or 12.2 $\mu$W (high intensity in Fig. 2 and Fig. 3). Fluorescence was detected using the Zeiss QUASAR detection unit. Image resolution was 768 × 450 pixels, with a pixel size of 0.23 $\mu$m. Sequential Z stacks separated by 0.5 $\mu$m were acquired with a time interval of 20 s between each frame, except for the export-recovery experiment in Fig. 1, in which we used 6.5 s.

### Image processing
Image analysis of live embryo movies was performed based on the protocol in[20,78], which included nuclear segmentation, spot segmentation, and tracking. In addition, the nuclear protein fluorescence of the Knirps repressor was calculated based on the protocol in[77]. The nuclear fluorescence of Knirps protein was calculated based on a nuclear mask generated from the MCP-mCherry channel. Knirps concentration for individual nuclei was extracted based on the integrated amount from maximum projection along the z-stack. The YFP background was calculated based on a control experiment and subsequently subtracted from the data.

### Predicting Knirps binding sites
To dissect Knirps binding to the _eve_ 4 + 6 enhancer, we used Patser[79] with already existing point weight matrices[80] to predict Knirps binding sites. The predicted binding sites with scores higher than 3.5 are shown in Supplementary Fig. 6.

### Compound-state Hidden Markov Model
To obtain the inference results shown in Fig. 4C, transcriptional traces were divided into 15 min-long segments. Each trace segment was then assigned to an inference group based on the average nuclear Knirps concentration over the course of its 15 min span. Trace segments with an average Knirps concentration of less than or equal to 4 arbitrary fluorescence units (au) were assigned to the "low" group and segments with a Knirps concentration greater than or equal to 6 au were assigned to the "high" group. Parameter estimates for each group were obtained by taking the average across no fewer than 21 separate bootstrap samples of the "high" and "low" trace segment groups. Each bootstrap sample contained a minimum of 6,027 and 10,000 time points for the high and low groups, respectively. Inference uncertainty was estimated by taking the standard deviation across these bootstrap replicates. We used a model with two burst states (OFF and ON) and an elongation time of 140 s (equal to seven time steps; see[57]).

### Input-output model parameter inference
We developed a simulation-based framework to infer the microscopic parameters that dictate how Knirps concentration regulates burst dynamics at the _eve_ 4 + 6 locus (Fig. 4F). This approach utilizes real Knirps concentration trends (Fig. 4D and E) from nuclei in our live imaging experiments to predict the burst frequency and burst duration in individual cells as a function of time using Supplementary Equations (1) and (2). These time-dependent burst parameter trends were then used to simulate populations of MS2 traces. By comparing predicted Knirps-dependent trends to our experimental results (Fig. 4G and H), we could assess how well a given set of microscopic parameters described our data. Using this procedure, we employed parameter sweeps and MCMC sampling to identify the best-fitting a set

of microscopic parameters. See Supplementary Note 2 and Supplementary Fig. 12 for additional details about our approach. See Supplementary Figs. 14 and 15 for additional input-output inference results.

## Thermodynamic binding model

We fit a simple equilibrium binding model (black curve in Fig. 4F) to the burst frequency vs. Knirps trend uncovered by our input-output inference (blue curve in Fig. 4F) to assess whether the observed frequency modulation was consistent with equilibrium repressor binding at the *eve*4 + 6 enhancer. The central assumption of this model is that that $k_{on}$ is inversely proportional to the number of Knirps molecules bound to the locus, such that

$$k_{on} = k_{on}^0 \left(1 - \frac{n_b}{N}\right), \tag{2}$$

where $k_{on}^0$ is the maximum burst frequency value (set to the $2.8\,\text{min}^{-1}$ value returned by MCMC inference), $n_b$ is the number of Knirps molecules bound, and $N$ is the total number of binding sites along the enhancer (set to 10 for the *eve* 4 + 6 enhancer; see Supplementary Fig. 6). Knirps-dependence enters into Equation (2) through $n_b$, which varies as a function of Knirps concentration. See Supplementary Note 3 for additional details.

## Reporting summary

Further information on research design is available in the Nature Portfolio Reporting Summary linked to this article.

## Data availability

All imaging data reported in this paper will be shared by the lead contact Hernan G. Garcia (hggarcia@berkeley.edu) upon request. The processed data that support the findings of this study are available in this paper's Github repository (https://github.com/GarciaLab/OptogeneticDissection). Source data for figures are provided with this paper. Source data are provided with this paper.

## Code availability

All code is available in this paper's Github repository (https://github.com/GarciaLab/OptogeneticDissection).

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

## Acknowledgements

We would like to thank Jack Bateman, Augusto Berrocal, Gary Karpen, Kirstin Meyer, Brandon Schlomann, Max Staller, Robert Tjian, Meghan Turner, and Orion Weiner for their comments on the manuscript. We thank all the Garcia Lab members for inspiring discussions. NCL was supported by NIH Genomics and Computational Biology training grant (5T32HG000047-18), the Howard Hughes Medical Institute, and DARPA under award number N66001-20-2-4033. HGG was supported by NIH R01 Awards R01GM139913 and R01GM152815, by the Koret-UC Berkeley–Tel Aviv University Initiative in Computational Biology and Bioinformatics, and by a Winkler Scholar Faculty Award. HGG is also a Chan Zuckerberg Biohub Investigator (Biohub–San Francisco).

## Author contributions

Conceptualization: J.Z., N.C.L., S.A., Y.J.K., H.G.G. Methodology: J.Z., N.C.L., S.A., Y.J.K., H.G.G. Resources: J.Z., N.C.L., S.A., Y.J.K., G.M., H.G.G. Investigation: J.Z., N.C.L., H.G.G. Visualization: J.Z., N.C.L., H.G.G. Funding acquisition: H.G.G. Project administration: H.G.G. Supervision: H.G.G. Writing - original draft: J.Z., N.C.L., H.G.G. Writing - review & editing: J.Z., N.C.L., S.A., Y.J.K., H.G.G.

## Competing interests

The authors declare no competing interests.
