## [Peer Review file · Nature Communications]

Optogenetic dissection of transcriptional repression in a multicellular organism

Corresponding Author: Dr Jiayi Zhao

Version 0:

Reviewer comments:

Reviewer #1

(Remarks to the Author)

Zhao et al. employ a combination of quantitative live imaging and optogenetics to explore the mechanisms of transcriptional repression in the early *Drosophila* embryo. Particular efforts focus on a classical gap repressor encoded by Knirps. Previous studies suggested that the Knirps repression gradient differentially regulates the even-skipped (*eve*) stripe 4+6 and stripe 3+7 enhancers. Here, the authors examine the consequences of rapid nuclear export of Knirps on the expression of a stripe 4+6 transgene. They present evidence for instantaneous changes in the bursting dynamics of the transgene upon optogenetic manipulation. The finding is interesting but I have a few questions and comments that should be considered in a revised manuscript:

1. The 4+6 enhancer requires relatively high concentrations of Knirps for repression (ie. formation of the posterior stripe 4 border and anterior stripe 6 border). Considerably lower concentrations of Knirps are required for the regulation of the 3+7 enhancer (posterior stripe 3 and anterior stripe 7). Have the authors looked at the 3+7 enhancer or predictions for the types of changes that might be expected in this type of analysis?
2. The Knirps expression pattern undergoes an anterior shift during the timeframe of this analysis. Have the authors considered their modeling in light of this shift?
3. The authors emphasize changes in burst frequency upon opto-manipulation of the Knirps repression gradient. Is it possible that Knirps also influences burst durations? In principle, if Kni was bound to the transgene during a burst, would this burst be completed or reduced in duration?
4. Does Knirps contribute to bursting dynamics, or does it simply serve to increase the probability that a subsequent burst will not occur?

Reviewer #2

(Remarks to the Author)

In this manuscript, Zhao and colleagues studied how transcription rate and bursting kinetics are regulated by a transcription factor (TF) in developing *Drosophila*. Employing an optogenetic system controlling the nuclear export of a zinc-finger repressor Knirps and two-color tagging, they were able to simultaneously record the repressor concentration and transcription dynamics in single cells. They found that Knirps downregulates its target gene even skipped (*eve*) in a switch-like manner and Knirps functions as the only repressor. It is shown that such strong repression is rapidly reversible and does not depend on its history. Using mathematical modeling and computational inference, they revealed that Knirps renders its negative regulation via reducing the burst frequency of *eve* transcription, which can be quantitatively explained by an equilibrium model of rapid Knirps binding.

This study provides an elegant framework to address how transcription factors shape gene expression in kinetic and molecular details. The experiments were beautifully designed and carefully carried out, and the data-driven mathematical modeling was performed on a solid basis. Simultaneous measurement of TF activity and the resulting transcription rate in single developing cells is the shining point of the work. However, it seems that the authors haven't explored the full potential of their data and models, for example the spatial changes in Knirps concentration, heterogeneity analyses and theoretical

generalization of the reversible repression mode of a TF. To reach the impact expected by the broad readership of Nature Communication, additional analyses and insights would be needed on some topics below.

Most conclusions of the current manuscript were obtained only from a narrow spatial region centered by the Knirps concentration peak (-2% to 2% window along the anterior-posterior axis), although a more extensive region was well measured (Figures S1-S3). It is not clear whether all main conclusions will still hold for cells beyond the 2% window. Further exploration of the whole data would make the results more solid and bring deeper insights.

1. In Figure S3, while the Knirps concentration varies (almost) symmetrically along its peak position (panel A), the transcription rate of *eve* shows clearly an asymmetric distribution (panel B). How do the authors comment on this? In addition, the maximum transcription rate is reached at position ~ 8% and time ~30 min, but the corresponding Knirps concentration is not the lowest. How can this be explained by the monotonic gene regulation function obtained in Figure 2D? It would also be nice to clarify why both TF concentration and transcription rate are zero in the far-left region (from -15%).
2. Can the input-output relation along the anterior-posterior axis (e.g., -15% to 15%) for a short time window be recapitalized by the same regulation curve in Figure 2D? And how does the curve look like when plotting all the spatial-temporal data points?
3. Is Knirps the sole repressor of *eve*, using optogenetic perturbation, in cells at other positions along the anterior-posterior axis?
4. In the export-recovery experiment, are the single-cell reactivation trajectories (Figure 3C), or the reactivation times, position dependent? Accordingly, does the recovery curve (Figure 3D) shift outside the 2% spatial window?
5. If the reactivation is totally memoryless, then the reactivation time is expected to be exponentially distributed, i.e., one-step process. But here we see a peaked distribution in Figure 3E, a hallmark for multi-step process. How to interpret this result? Could there be a weak memory due to multi-step reactivation of the promoter state, even without epigenetic regulation?
6. There is no description of the red dots in the legend of Figure 3F, which appears to be means of binned data. To demonstrate the independence of reactivation time on dwell time in repressed state, it is ideal to perform a statistical test for the trend between the two quantities, instead of using binned statistics.

One highlight of the work is spatiotemporal measurements in single cells in living embryos, but quantification of transcription heterogeneity or noise seems to be not sufficient.

7. How large are the noises of Knirps abundance and transcription rate in cells within an embryo? Can the heterogeneity of *eve* transcription rate be explained by that of the Knirps level?
8. How does the transcriptional noise vary with time and location?
9. How do the reversible and switch-like features of Knirps action contribute to the control of transcriptional noise of its target? Is reducing the burst frequency beneficial to limit the gene expression noise compared to other modulation modes?

There is still potential for exploring the results of inferring transcriptional bursting kinetics.

9. What cells in terms of spatial location were used to perform the inference on parameters of transcriptional bursting? In whole measured region or only limited window (as in Figure 2D)?
10. It is of great interest that the repression of the actual transcription rate and burst frequency show different sensitivity (comparing Figure 2D and 7G to Figure 7F). Thus, transcription will persist (with half-saturation point ~ 6 au in Figure 2D and 7G) even after the decline of burst frequency (with half-saturation point ~ 3.5 au in Figure 7F). Why is this so? And is there any biological significance for this?
11. The authors showed that a rapid TF binding model can explain the modulation of the burst frequency by Knirps. Combining with point 10 above, it implies that transcription would only start to slow down when the repressor binding already reaches saturation, enabling somewhat inefficient control. This might link to a paper by Li et al (2018 Cell Systems), in which modulating the speed of the gene-state cycle can shift the sensitivity of transcription without change in TF binding. Could this be something relevant or do the authors have other explanations?

(Remarks to the Author)

This beautiful work by the Garcia team combines a powerful array of live imaging techniques (quantitative live-imaging of protein-concentration, RNA expression, and real-time protein perturbations) to advance our understanding of the critical timescales of transcriptional regulation and quantitative relationships between repressor levels and transcription. I think it is an important advance deserving publication.

My main concerns are described in detail below, and I believe can be addressed without further experimentation.

1) Introduction

I felt the Introduction significantly under represents the authors achievements. By combining in a single sentence quantification and manipulation of proteins I think the key innovation of the work is lost.

As I am certain the authors are aware, a decent amount of work has been done to *measure* transcriptional dynamics in (mostly looking at nascent transcripts) in both cell culture and live Drosophila embryos. I'm not sure it's fair today to even say the animal models lag behind the cell culture when it comes to the MS2 imaging, given all the progress. It is more fair to say less has been done on the protein imaging. But all this misses what to me is one of the most valuable innovations of this work, which is high speed control/manipulation of transcription factors. Clear demonstrations of control (as opposed to measurement) of transcriptional regulation on the sub-minute time scale is lacking in cell culture models as much as in live animal models.

The importance of this innovation cannot be overstated. There is a world of difference between hour-long degenon experiments and sub-minute optical control (especially when it comes to testing reversibility). And there aren't even many published works, to my knowledge, combine degenons in real time to study transcriptional kinetics or chromatin kinetics. (Beautiful recent work by the Hansen lab and Giorgetti labs for instance use degenons and live chromatin imaging, but only as replacement for lethal mutations, not as a kinetic tool – Gabriele 2022, Mach 2022). I think the utility of rapid and precise manipulation could be better introduced and better distinguished from the important, but much more developed arts of live single-molecule tracking. These go beyond the implications for the interesting but more focused question about the duration of repression.

The jump to irreversible vs. reversible also belies the finesse of the approach that can measure sub-minute. There are serious biological implications to reversibility in <1 min vs. reversible repression in ~10 minutes. Recent genetic experiments have made it clear conserved enhancer (as for Scr) appear to function primarily to avoid these ~ minute reviews.

To me much of the discussion about irreversibility of repression hinges on our understanding of epigenetic inheritance, and I don't think most of the field would be surprised to see this cannot be stably (irreversibly) established within the scale of ~30 min time scales assayed here. What to me is more exciting is that the reversibility can be measured at all at these time scales. Other authors (e.g. Bintu...Elowitz 2016, Ref 13) have looked at the reversibility in mammalian (immortalized) cell culture at the time resolution of days, not minutes (not at the single molecule scale). Bintu has shown that the difference between 1 day and 5 days of induced repression have different degrees of reversibility. Thus, the big question is not so much the qualitative one of what is reversible vs. irreversible, but really knowing the quantitative time scales for these with resolutions of minutes, not days. We can do a much better job testing/eliminating potential models of gene regulation with this revolutionary sub-minute temporal control introduced here.

2) Figure 2

I am confused by the numbers on the graphs on Fig 2D vs. Fig 2E. Is a different normalization being used? It appears the 3 examples in E all show transcription shutting down when the relative Kni levels reach 10 au (which continues to rise to 16 au). However, in 2D the max Kni levels are 10 au, and repression is at half max at 6 au (which is the level at which the graphs on 2E start). Similarly the transcription rates go from 0 to 20 in D and 0 to 1.2 in E. Either I'm reading this all wrong or there is some unnecessary discrepancy in the normalization. I appreciate it's all "arbitrary units" but it would be nice if the combined trace examples were on the same scale as the single cell examples.

Version 1:

Reviewer comments:

Reviewer #1

(Remarks to the Author)

The revised manuscript by Zhao et al. nicely incorporates my previous concerns and suggestions, and is now suitable for publication. I thank the authors for their thoughtful comments.

Reviewer #2

(Remarks to the Author)

The authors have fully addressed my comments. I believe the revised manuscript is an important contribution to gene regulation community. Congrats!

Reviewer #3

(Remarks to the Author)

The authors have addressed my concerns. In my opinion, they have also adequately addressed the reasonable concerns of the other reviewers. In particular, I find the author's case that examination of the 3+7 enhancer is interesting but not necessary to support the claims of the current article and an interesting project for future work.

Thank you again for submitting your manuscript "Optogenetic dissection of transcriptional repression in a multicellular organism" to Nature Communications. We have now received reports from 3 reviewers and, on the basis of their comments, we have decided to invite a revision of your work for further consideration in our journal. Your revision should address all the points raised by our reviewers (see their reports below).

When resubmitting, you must provide a point-by-point response to the reviewers' comments. Please show all changes in the manuscript text file with track changes or colour highlighting. If you are unable to address specific reviewer requests or find any points invalid, please explain why in the point-by-point response.

Author response: We thank the reviewers for their insightful comments and suggestions on our manuscript. We tried our best to address the reviewers' concerns and believe that their comments have significantly improved our manuscript. Please see below for our detailed point-by-point responses to reviewer comments (in blue). All corresponding changes to our manuscript are marked in red.

REVIEWER COMMENTS

Reviewer #1 (Remarks to the Author):

Zhao et al. employ a combination of quantitative live imaging and optogenetics to explore the mechanisms of transcriptional repression in the early *Drosophila* embryo. Particular efforts focus on a classical gap repressor encoded by Knirps. Previous studies suggested that the Knirps repression gradient differentially regulates the even-skipped (*eve*) stripe 4+6 and stripe 3+7 enhancers. Here, the authors examine the consequences of rapid nuclear export of Knirps on the expression of a stripe 4+6 transgene. They present evidence for instantaneous changes in the bursting dynamics of the transgene upon optogenetic manipulation. The finding is interesting but I have a few questions and comments that should be considered in a revised manuscript:

Author response: We thank the reviewer for the thoughtful review and valuable suggestions. We greatly appreciate the time and effort the reviewer has invested in carefully assessing our

work and providing insightful feedback on potential areas for further investigation. In the following point-by-point response, we address each of your concerns and recommendations. Where feasible, we have endeavored to expand the scope of our manuscript to incorporate the suggestions. In cases where we believe certain proposed directions extend beyond the current study's scope, we have aimed to provide a comprehensive rationale for our perspective and to provide that same rationale within the paper. Please find below our detailed responses to the reviewer's comments. We hope that our revisions and explanations adequately address the reviewer's concerns and enhance the overall quality of our manuscript.

1. The 4+6 enhancer requires relatively high concentrations of Knirps for repression (ie. formation of the posterior stripe 4 border and anterior stripe 6 border). Considerably lower concentrations of Knirps are required for the regulation of the 3+7 enhancer (posterior stripe 3 and anterior stripe 7). Have the authors looked at the 3+7 enhancer or predictions for the types of changes that might be expected in this type of analysis?

Author response: We agree with the reviewer that the contrasting sensitivities of 4+6 and 3+7 enhancers to Knirps concentration could make for an interesting comparison. The proposal to extend our research to investigate the 3+7 enhancer, which is sensitive to relatively lower concentrations of Knirps, indeed opens up intriguing avenues to explore the regulatory mechanisms further.

While we acknowledge the potential insights such an exploration might yield, our present experimental setup does not permit the full export of Knirps from the nucleus. Indeed, as detailed in Figure 3D of our manuscript, the concentration of Knirps in the nucleus is reduced only by $\approx 50\%$ upon light exposure. The *eve* 3+7 enhancer is highly sensitive to Knirps repression, and a relatively low concentration of Knirps can repress *eve* 3+7 enhancer as previously shown (Struffi et al., *Development*, 2004). Due to this limitation, we anticipate difficulties in inducing meaningful perturbations in the *eve* 3+7 enhancer, since we likely cannot drive repressor levels low enough to probe the regulatory logic of the 3+7 enhancer.

However, we view the exploration of *eve* 3+7 enhancer and similar enhancers as a key point of emphasis for future research projects. We are actively working on improving our optogenetic framework to overcome the current limitations and expand our research to cover a broader range of enhancer elements. We appreciate the reviewer's thoughtful suggestion and have

added mention of how better optogenetics modulation could help dissect enhancers that are more sensitive to TF concentrations in the Discussion section of our manuscript (lines 343-345).

2. The Knirps expression pattern undergoes an anterior shift during the timeframe of this analysis. Have the authors considered their modeling in light of this shift?

Author Response: We thank the reviewer for highlighting the dynamic nature of the Knirps expression pattern and its implications for our analysis. Indeed, the shifting center of the Knirps domain during the time frame of this study presents a noteworthy challenge. Our modeling approach (depicted in Figure 4D-H) overcomes this challenge by incorporating real-time single-cell measurements of changing Knirps concentrations. This allows us to track and account for changes in Knirps concentration and *eve* stripe 4+6 expression at the level of individual cells on a minute-by-minute basis. This granularity in our model ensures that any shift, including the anterior movement of the Knirps domain, is effectively captured and reflected in our results. We refer the reviewer to Section 2 of our Supplementary Text and Figure S12 for further details about our modeling approach.

For completeness, we quantified the movement of the Knirps domain. Our analysis, now presented as new Figure S4 in the supplemental materials, reveals that the center of the Knirps domain shifts approximately 2% toward the anterior during nuclear cycle 14, consistent with previous measurements (Jaeger et al., *Nature*, 2014).

Figure S4 (new): Temporal dynamics of the Knirps expression center domain. The graph shows the change in the position of the center of the Knirps expression domain as a function of time. Our analysis reveals that the center of the Knirps domain shifts approximately 2% toward the anterior during nuclear cycle 14, consistent with previous measurements.

3. The authors emphasize changes in burst frequency upon opto-manipulation of the Knirps repression gradient. Is it possible that Knirps also influences burst durations? In principle, if Kni was bound to the transgene during a burst, would this burst be completed or reduced in duration?

Author response: We thank the reviewer for raising this important question. Dissecting where repressors act within the kinetics of the transcriptional burst cycle was one of our central motivations in pursuing this study. Our burst inference results (left panel of Figure 4C) indicate quite definitively that Knirps represses *eve* 4+6 by decreasing the frequency of transcriptional bursts. From a physical perspective, we interpret this to mean that Knirps acts upon the gene locus *in between* bursts, which suggests that Knirps binding during a burst would not have any effect in decreasing its duration. We have updated the discussion to emphasize this important point (see lines 323-324 of the revised manuscript). Overall, we cannot completely rule out an effect of Knirps binding on extant bursts, as the reviewer mentions, but our results suggest that such a mechanism is not a significant driver of transcriptional control by Knirps.

4. Does Knirps contribute to bursting dynamics, or does it simply serve to increase the probability that a subsequent burst will not occur?

Author response: We thank the reviewer for the insightful question regarding the role of Knirps in burst dynamics. However, we found the reviewer's question, as stated, challenging to parse. One interpretation is that the reviewer is asking whether Knirps is responsible for generating the bursts in the first place, or whether it merely acts to modulate already existing bursts dictated by other molecular players. Our data suggests that the bursts exist even in the absence of Knirps. For example, the center of every stripe of the *even-skipped* pattern exhibits robust burst-like expression (Berrocal et al., *eLife* 2020), despite the fact that repressor levels tend to be vanishingly small at these locations.

Alternatively, the reviewer could be asking whether Knirps repression is effectively binary, in the sense that Knirps binding at a locus can either abolish bursting altogether or leave it unchanged. In this case, there would be no measurable concentration-dependent modulation of burst dynamics at individual gene loci. Instead, there would be a sharp Knirps-dependent decrease in the fraction of active nuclei. However, while possible, we believe that our qualitative and quantitative findings rule out this hypothesis and, instead, indicate that Knirps modulates burst dynamics in a concentration-dependent fashion. Consider our activity traces displayed in the middle panel of Figure 4B. Here, bursting is observed to persist even as levels of Knirps escalate, but the bursts clearly spread out over time. This observable pattern aligns with the understanding that Knirps is extending the inactive intervals between the bursts, thus directly influencing the bursting dynamics. Our modeling results provide quantitative support for this stance, demonstrating that increasing Knirps concentration systematically decreases the burst frequency, as portrayed in Figure 4F.

Reviewer #2 (Remarks to the Author):

In this manuscript, Zhao and colleagues studied how transcription rate and bursting kinetics are regulated by a transcription factor (TF) in developing *Drosophila*. Employing an optogenetic system controlling the nuclear export of a zinc-finger repressor Knirps and two-color tagging, they were able to simultaneously record the repressor concentration and transcription dynamics in single cells. They found that Knirps downregulates its target gene *even-skipped* (*eve*) in a switch-like manner and Knirps functions as the only repressor. It is shown that such strong repression is rapidly reversible and does not depend on its history. Using mathematical modeling and computational inference, they revealed that Knirps renders its negative regulation via reducing the burst frequency of *eve* transcription, which can be quantitatively explained by an equilibrium model of rapid Knirps binding.

This study provides an elegant framework to address how transcription factors shape gene expression in kinetic and molecular details. The experiments were beautifully designed and carefully carried out, and the data-driven mathematical modeling was performed on a solid basis. Simultaneous measurement of TF activity and the resulting transcription rate in single developing cells is the shining point of the work. However, it seems that the authors haven't explored the full potential of their data and models, for example the spatial changes in Knirps concentration, heterogeneity analyses and theoretical generalization of the reversible repression mode of a TF. To reach the impact expected by the broad readership of *Nature Communication*, additional analyses and insights would be needed on some topics below.

Author response: We thank the reviewer for their thoughtful assessment of our work and for their insightful feedback regarding further avenues of inquiry. Below, we provide point-by-point responses to each of the reviewer's concerns and suggestions. Here, we briefly summarize key steps that we have taken in response to your feedback.

1. We have extended two key analyses to include a broader spatial region within the developing embryo (-6 to 6% AP from the center of the Knirps domain), confirming that our core conclusions regarding the sharpness (Figure S6) of Knirps repression and the speed of *eve* 4+6 reactivation (Figure S10) remain unchanged.
2. We have introduced new plots (Figure S3 C&D) and calculations (SI Section 5) to quantify the noise levels in our data and to probe the potential benefits (or lack thereof) of frequency-based repression from the perspective of transcriptional noise.
3. We have added a new supplementary figure (Figure S14) to shed light on how Knirps-dependent trends in k_{on} and (to a lesser extent) k_{off} interact to produce the transcription rate vs. Knirps repression curve shown in Figure 4G.
4. We have substantially revised the discussion to incorporate key points from the reviewer regarding the potential for using our optogenetic system to dissect the role of heterogeneity and noise in transcriptional regulation.

We believe that these revisions have significantly improved our work, and we thank the reviewer for their insightful suggestions.

Most conclusions of the current manuscript were obtained only from a narrow spatial region centered by the Knirps concentration peak (-2% to 2% window along the anterior-posterior axis), although a more extensive region was well measured (Figures S1-S3). It is not clear whether all main conclusions will still hold for cells beyond the 2% window. Further exploration of the whole data would make the results more solid and bring deeper insights.

Author response: Briefly, regarding our decision to focus on the center of the Knirps domain, we chose to do so because this is the only region within the fly embryo where Knirps could reasonably be expected to function as the dominant regulator of *eve* 4+6 activity. Specifically, as briefly described in lines 102-104, we reasoned that expanding the region beyond this point would introduce the influence from other repressors, such as Hunchback, Giant, and Krüppel, significantly complicating our analysis without shedding further light on the mechanism of Knirps repression. Nonetheless, we take the reviewer's concern seriously and have re-run

several key analyses on a broader region from -6 to 6% of the domain center. As detailed in responses 2-4 below, these efforts confirmed that our original conclusions about both the sharpness of Knirps repression pattern and the speed of *eve* 4+6 reactivation remain unchanged for this broader spatial region.

1. In Figure S3, while the Knirps concentration varies (almost) symmetrically along its peak position (panel A), the transcription rate of *eve* shows clearly an asymmetric distribution (panel B). How do the authors comment on this? In addition, the maximum transcription rate is reached at position $\sim 8\%$ and time ~ 30 min, but the corresponding Knirps concentration is not the lowest. How can this be explained by the monotonic gene regulation function obtained in Figure 2D? It would also be nice to clarify why both TF concentration and transcription rate are zero in the far-left region (from -15%).

Author response: Thank you for raising this point. Indeed, the asymmetry in the transcription rate of *eve* cannot be solely accounted for by the monotonic model showcased in Figure 2D. This is because the regulatory logic shaping *eve* 4+6 activity becomes more complex as one moves farther from the center of the Knirps domain. In particular, while our study emphasizes the role of Knirps as the chief repressor for the posterior and anterior regions of stripes 4 and 6, respectively, it is known the anterior boundary of stripe 4 and the posterior boundary of stripe 6 form as a result of Hunchback repressor rather than Knirps repressor (Samee et al., *Cell Reports*, 2017), which is why the transcription rates are zero in the far-left region indicated by the reviewer. We have clarified this aspect in the revised manuscript, ensuring clarity for readers (see lines 102-104 of the revised manuscript).

2. Can the input-output relation along the anterior-posterior axis (e.g., -15% to 15%) for a short time window be recapitalized by the same regulation curved in Figure 2D? And how does the curve look like when plotting all the spatial-temporal data points?

Author response: We thank the reviewer for the question regarding the input-output relation along the anterior-posterior (AP) axis in our study. We have generated a new figure, Figure S6, in which we observed that the input-output relationship varies across different AP positions, distinguishing it from the pattern depicted in Figure 2D. In Figure S6, we have demonstrated that while the sharpness of the relationship remains consistent, a shift in the dissociation constant (K_d) is evident across various positions along the anterior-posterior axis. We

hypothesize that this spatial dependence might be attributed to the influence of specific activators or repressors interacting with enhancer elements, which lead to differential activation levels along the AP axis, causing the observed shifts in K_d . Given these considerations, and to minimize the influence from other transcriptional repressors, such as Hunchback (see response 1), we have chosen to focus our analysis regarding repressor activity on the narrow range of -2% to 2% and not include all spatial-temporal data points in a single plot. We believe aggregating all data points might mask the subtle yet significant differences in input-output relationships at different anterior-posterior coordinates.

3. Is Knirps the sole repressor of *eve*, using optogenetic perturbation, in cells at other positions along the anterior-posterior axis?

Author response: We thank the reviewer for the question regarding the role of Knirps in repressing *eve* expression across different positions along the anterior-posterior axis. As demonstrated in Figure S10, we analyzed the reactivation dynamics in cells positioned at -6% to -2% and 2% to 6% along this axis. Our findings show that in these adjacent regions, the reactivation of *eve* stripe 4+6 following optogenetic perturbation occurs completely and within a similar time frame compared to the -2% to 2% region shown in Figure 3D. This consistency in reactivation dynamics strongly suggests that Knirps functions as the sole repressor of *eve* in these specific regions.

However, It is important to note that other repressors also contribute to the formation of *eve* stripes 4 and 6. Specifically, as detailed in response 1, the Hunchback protein acts as a crucial repressor that helps define the anterior edge of stripe 4 and the posterior edge of stripe 6 (Samee et al., *Cell Reports*, 2017).

4. In the export-recovery experiment, are the single-cell reactivation trajectories (Figure 3C), or the reactivation times, position dependent? Accordingly, does the recovery curve (Figure 3D) shift outside the 2% spatial window?

Author response: We find no evidence for positional dependence in transcriptional reactivation following Knirps export. The new Figure S10 shows recovery curves for regions directly to the anterior (-6% to -2%) and posterior (2% to 6%) of the center of the Knirps domain. We find that these recovery curves exhibit comparable reactivation dynamics to the 1-4 minute timescale

shown in Figure 3D. This observation indicates that, within these broader spatial windows, the reactivation dynamics are not significantly dependent on the positional context.

Regarding the assessment of single-cell reactivation times, we encountered a technical limitation: there was an insufficient number of cells that exhibited complete repression in these positions to make representative measurements of reactivation times at the single-cell level. Consequently, we opted to focus on the “fraction on” metric, as this provides a more reliable and representative comparison of reactivation dynamics under the constraints of our experimental setup. This approach is detailed in Figure S10, where we compare the fraction of cells reactivated over time across different positions. We found that, within the constraints of our current data, the reactivation dynamics do not appear to be positionally dependent. As has been noted in responses 1 and 2; lines 102-104, these conclusions would not be expected to apply in other regions of the embryo, where additional repressors such as Hunchback may play important roles in regulating *eve* 4+6.

5. If the reactivation is totally memoryless, then the reactivation time is expected to be exponentially distributed, i.e., one-step process. But here we see a peaked distribution in Figure 3E, a hallmark for multi-step process. How to interpret this result? Could there be a weak memory due to multi-step reactivation of the promoter state, even without epigenetic regulation?

Author response: We thank the reviewer for raising this point. The reviewer is right to note that a memoryless single-step process would generally be expected to produce exponentially distributed reactivation times; however, two factors in our system cause the observed distribution to deviate from this expectation.

First, our live imaging experiments have a finite detection threshold, which means that there is some delay between when a gene locus *actually* first turns on and when we can first detect this event. This can be seen by comparing the observed reactivation CDF with the “true” reactivation CDF—i.e. the CDF assuming no detection limit—predicted by our model in Figure S15. Second, the k_{on} rate increases significantly over time (blue curve in Figure S15). As a result, even absent the detection threshold effect, the reactivation times would not be expected to follow a simple exponential distribution, since they reflect the action of a time-varying kinetic rate. We have substantially revised the discussion to emphasize the challenges that these

experimental factors present in our efforts to model the transcriptional system (lines 311-318 of the revised manuscript).

6. There is no description of the red dots in the legend of Figure 3F, which appears to be means of binned data. To demonstrate the independence of reactivation time on dwell time in repressed state, it is ideal to perform a statistical test for the trend between the two quantities, instead of using binned statistics.

Author response: As suggested by the reviewer, we have conducted a more rigorous statistical test to assess the relationship between OFF duration and reactivation time depicted in Figure 3F. Specifically, we have fit a linear regression model to the data to assess the dependence of reactivation time on the duration of the repressed state. The results of this analysis are now included in the revised figure. Our analysis revealed that the slope coefficient of the regression model has a p-value of 0.50. Since this value is significantly greater than the conventional threshold for statistical significance ($p = 0.05$), it indicates that there is no statistically significant trend between reactivation time and dwell time in the repressed state. This finding supports the notion that reactivation time is independent of the duration of the repressed state. Additionally, we have updated the legend of Figure 3F to include a clear description of the red dots, representing the means of the binned data. This addition will aid in better understanding and interpretation of the figure.

Overall, this revised analysis and the updated figure legend should more effectively demonstrate the independence of reactivation time from duration in the repressed state, as the reviewer suggested.

One highlight of the work is spatiotemporal measurements in single cells in living embryos, but quantification of transcription heterogeneity or noise seems to be not sufficient.

Author response: We agree with the reviewer that investigating the role of transcriptional heterogeneity in gene regulation represents a fascinating research direction. We have conducted several new analyses in response to reviewer questions on this front. Overall, our preliminary calculations indicate that intrinsic variability from bursting—not extrinsic variability in Knirps—is the dominant source of noise in *eve* 4+6 transcription.

We have introduced new figure panels (Figure S3 C&D) and a new SI Section (Section 5) to reflect these new analyses. However, we believe that a large-scale study of transcriptional and protein heterogeneity lies beyond the scope of the present work. This is chiefly because measuring noise within live imaging experiments is inherently difficult, and further refinements to our experimental system would be required to disentangle technical and biological sources of noise in our transcriptional and protein measurements. Additionally, the fact that Knirps concentrations (and therefore bursting) never reach a steady state presents theoretical challenges that merit more extensive treatment than can be offered in the context of the present work.

While not insurmountable, we hope the reviewer will agree that addressing these significant technical and conceptual efforts would constitute an entire, separate body of work.

7. How large are the noises of Knirps abundance and transcription rate in cells within an embryo? Can the heterogeneity of *eve* transcription rate be explained by that of the Knirps level?

Author response: We have added new panels to Figure S3 (C & D) to quantify the variability of Knirps concentrations and *eve* 4+6 transcription as a function of space and time. Overall, the presence of transcriptional bursts in our data provides a strong indication that there will be a large amount of intrinsic variation in the transcription rate that cannot be explained by extrinsic variability in Knirps concentrations. Below, we outline a series of simple calculations that we conducted to confirm this expectation. To cut a long story short, our findings indicate that intrinsic variability from bursting (coefficient of variation (CV) = 0.91) is predicted to be several times larger than extrinsic variation from noise in Knirps levels (CV=0.13).

To obtain these numbers, we confined our analysis to the wild-type embryos (i.e., embryos not exposed to optogenetic manipulation) and focused on the first 20 minutes of nuclear cycle 14 (when most *eve* 4+6 loci are still relatively active). At each time point during this period, we calculated the nucleus-to-nucleus variability in both input Knirps levels and output *eve* 4+6 transcription rates, using MS2 intensity as a proxy for transcription rate. We found that the transcription rate had a CV of 0.88 during this period, and Knirps levels had a CV of 0.24.

To determine whether the observed input noise in Knirps can explain the output variation in *eve* 4+6 activity, we need an analytic expression that can relate the transcription rate to Knirps concentration. We can express the rate of transcription as a function of k_{on} , k_{off} , and r :

$$\text{rate} = r * k_{on} / (k_{on} + k_{off}),$$

Equations S1 and S2 in our manuscript give k_{on} and k_{off} as a function of Knirps concentration. Plugging these expressions into the above equation thus gives us an analytic link between repressor concentration and the rate of transcription. Using linear noise propagation, it is then possible to predict how much transcriptional noise should result from observed noise in Knirps levels (CV=0.24). Using this approach, we found that variation in Knirps concentrations alone would lead to a transcriptional CV of only 0.13; far lower than the observed value of CV=0.88.

We next examined how much noise transcriptional bursting was predicted to contribute. Specifically, we used the values of k_{on} , k_{off} , and r returned by our model fits to get predictions for the *intrinsic* noise levels from bursting during this same 20-minute period. To do this, we used theoretical expressions for burst-dependent variance in transcription from Lammers et al (PNAS, 2023). This gave a predicted CV of 0.90, far closer to the actual value we observed.

We stress that these measurements and calculations are preliminary in nature, and that substantial experimental and theoretical work would be needed to obtain reliable noise estimates from our data. Nonetheless, we believe that the above calculations suggest that transcriptional bursting, not Knirps variability, constitutes the dominant source of noise in the transcription rate. In addition to the analysis outlined above, we have updated the Discussion to point out that the analysis of transcriptional heterogeneity constitutes an important next step in the development of the theoretical and experimental approaches presented in this work (see lines 343-352 of the revised manuscript).

8. How does the transcriptional noise vary with time and location?

Author response: We agree that it is of interest to track noise levels across time and space. We have updated Figure S3 to include panels that show how the coefficient of variation in Knirps levels (Figure S3C) and *eve* 4+6 transcription (Figure S3D) vary as a function of anterior-posterior position and minutes into nuclear cycle 14. As discussed above, we find that transcriptional noise levels are substantially higher than noise levels in input Knirps concentration.

Transcriptional noise levels ramp up in the latter half of nuclear cycle 14 (25+ minutes) in the region between stripes 4+6. In contrast, Knirps noise levels remain low in this region over the full period of observation.

9. How do the reversible and switch-like features of Knirps action contribute to the control of transcriptional noise of its target? Is reducing the burst frequency beneficial to limit the gene expression noise compared to other modulation modes?

Author response: This is an interesting thought. It is important to note that the reversible nature of Knirps repression was only revealed via optogenetic manipulations that pushed the system away from its normal trajectory. Consequently, it is unlikely that the reversibility itself plays a role in endogenous transcriptional control. Instead, we believe that it provides a useful window into the nature of the underlying molecular processes that drive Knirps repression.

With respect to the second part of the reviewer's question, we absolutely agree that it is of interest to examine how the down-regulation of the burst frequency impacts transcriptional noise. We have added a new SI section (Supplementary Text Section 5) that uses simple analytical arguments to compare burst frequency regulation to other modulation modes. Perhaps surprisingly, we find that the reduction of the burst frequency is actually predicted to be a suboptimal regulatory strategy from a noise perspective relative to burst amplitude- or burst duration-based repression.

We direct the Reviewer to the new SI section for further details. Much more work will be required to examine the interplay between regulatory strategy and transcriptional noise within the complex and dynamic context of the developing fly embryo, but we believe that the initial calculations we have added provide an interesting point of departure. As noted above, we have also updated the Discussion in the main text to emphasize the Reviewer's point that our optogenetic platform can serve as a platform for investigating the noise characteristics of gene regulation (lines 343-352 of the revised manuscript).

There is still potential for exploring the results of inferring transcriptional bursting kinetics.

Author response: We appreciate the reviewer's thoughtful analysis of our burst inference results. In particular, as detailed below, we believe that the disjunction in Knirps sensitivities

between (i) the k_{on} repression curve in Figure 4F and (ii) the transcription rate curve in Figure 4G points to interesting limitations in burst frequency-mediated repression. Taken together with our new analysis in SI Section 5 showing that frequency-based repression is also suboptimal from a noise perspective, this result raises fundamental questions about *why* the system is as it is. We believe that this line of questioning will provide a strong foundation for future work (lines 350-352).

9. What cells in terms of spatial location were used to perform the inference on parameters of transcriptional bursting? In whole measured region or only limited window (as in Figure 2D)?

Author response: We thank the reviewer for raising this point. The inference on the parameters of transcriptional bursting was specifically carried out using cells located within the spatial window of $\pm 2\%$ along the anterior-posterior axis from the center of the *Knirps* domain. As discussed in our response to the reviewer's first comment above, our decision to focus on this particular window is because the regulatory dynamics of *eve* stripes 4 and 6 are much more complicated outside of this region. We have now further elucidated this aspect in the manuscript, enhancing clarity on our chosen approach and the underlying rationale for such a choice (see lines 205-206 of the revised manuscript).

10. It is of great interest that the repression of the actual transcription rate and burst frequency show different sensitivity (comparing Figure 2D and 7G to Figure 7F). Thus, transcription will persist (with half-saturation point ~ 6 au in Figure 2D and 7G) even after the decline of burst frequency (with half-saturation point ~ 3.5 au in Figure 7F). Why is this so? And is there any biological significance for this?

Author response: We agree that the difference in *Knirps* sensitivity for the burst frequency and the transcription rate is striking. This can be explained by our finding that the burst duration actually increases somewhat as *Knirps* concentration increases (Figure 4C, S11C). This acts to offset the drop in burst frequency, thereby increasing the half-max point for the transcription rate vs. *Knirps* curve. We have added a new supplementary figure (Figure S14) to illustrate this point. One obvious question that this raises is why the model should infer such a sharp and sensitive response curve for the burst frequency rather than something that is in closer accord with the observed trend in the transcription rate. The answer to this is that the burst frequency *must* have a half-max point around 3.5-4au in order to explain the rapid reactivation of

transcription when we export Knirps from the nucleus (notice that Knirps levels remain around 4au even after export; Figure 3D).

Thus, while the burst duration does not directly contribute to the repression of *eve* 4+6, the moderate Knirps-dependent increase we observe appears to be *necessary* in order to explain the trends shown in Figure 4G & H. One way to think about this is that both the burst duration and the burst frequency impact the sensitivity of repression, but only burst frequency impacts the sensitivity of reactivation. Thus, our model is responding to real asymmetries between repression and reactivation in the underlying data. We believe that further investigating this difference represents a promising avenue for future work. We have added lines 268-270 to the main text to draw the reader's attention to this important point.

11. The authors showed that a rapid TF binding model can explain the modulation of the burst frequency by Knirps. Combining with point 10 above, it implies that transcription would only start to slow down when the repressor binding already reaches saturation, enabling somewhat inefficient control. This might link to a paper by Li et al (2018 Cell Systems), in which modulating the speed of the gene-state cycle can shift the sensitivity of transcription without change in TF binding. Could this be something relevant or do the authors have other explanations?

Author response: This is an interesting observation. We agree that the offset between the activation and burst frequency repression trends noted by the reviewer in comment 10 above would imply that *eve* 4+6 activity persists to some degree even when repressor binding is close to saturation. However, as noted above, this is due to an increase in burst duration that is coincident with a decrease in burst frequency. In practice, what this means is that, while an extant burst might persist for some period of time, even with near-saturating Knirps binding, it will eventually switch into the OFF state. At this point Knirps binding acts to maintain it in this state.

We agree that this repressive mechanism seems suboptimal; repressing the burst duration would be a much more direct approach. This could reflect countervailing constraints that favor burst frequency, or it may simply be the case that the observed mechanism is “good enough” for the developmental system to function within its constraints.

Reviewer #3 (Remarks to the Author):

This beautiful work by the Garcia team combines a powerful array of live imaging techniques (quantitative live-imaging of protein-concentration, RNA expression, and real-time protein perturbations) to advance our understanding of the critical timescales of transcriptional regulation and quantitative relationships between repressor levels and transcription. I think it is an important advance deserving publication.

Author response: We are grateful for the reviewer's generous and encouraging feedback on our study. We're thrilled to hear that the combination of live imaging techniques utilized in our research is recognized as a significant contribution to enhancing understanding in the realm of transcriptional regulation.

My main concerns are described in detail below, and I believe can be addressed without further experimentation.

1) Introduction

I felt the Introduction significantly under represents the authors achievements. By combining in a single sentence quantification and manipulation of proteins I think the key innovation of the work is lost.

As I am certain the authors are aware, a decent amount of work has been done to *measure* transcriptional dynamics in (mostly looking at nascent transcripts) in both cell culture and live Drosophila embryos. I'm not sure it's fair today to even say the animal models lag behind the cell culture when it comes to the MS2 imaging, given all the progress. It is more fair to say less has been done on the protein imaging. But all this misses what to me is one of the most valuable innovations of this work, which is high speed control/manipulation of transcription factors. Clear demonstrations of control (as opposed to measurement) of transcriptional regulation on the sub-minute time scale is lacking in cell culture models as much as in live animal models.

The importance of this innovation cannot be overstated. There is a world of difference between hour-long degron experiments and sub-minute optical control (especially when it comes to testing reversibility). And there aren't even many published works, to my knowledge, combine degrons in real time to study transcriptional kinetics or chromatin kinetics. (Beautiful recent work by the Hansen lab and Giorgetti labs for instance use degrons and live chromatin imaging, but only as replacement for lethal mutations, not as a kinetic tool – Gabriele 2022, Mach 2022). I think the utility of rapid and precise manipulation could be better introduced and better distinguished from the important, but much more developed arts of live single-molecule

tracking. These go beyond the implications for the interesting but more focused question about the duration of repression.

Author response: We thank the reviewer for their kind and perceptive feedback. We wholeheartedly agree that the introduction could do a better job of highlighting the key innovations of this work and have revised it substantially in light of the reviewer's suggestions. In particular, we have made the following changes:

- We have emphasized the crucial distinction between quantification and manipulation of proteins, and we have revised the introduction to ensure this innovation is highlighted adequately.
- We have removed the distinction between progress in using MS2 in cell culture vs. multicellular organisms.
- We have emphasized our contribution in offering high-speed control/manipulation of transcription factors, since we agree that this stands out as a pivotal innovation in this field, offering a nuanced understanding of transcriptional regulation at a sub-minute time scale—a dimension scarcely explored in both live animal and cell culture models.
- We have drawn an explicit distinction between our ability to directly manipulate TF concentrations and degron-based experiments which do not permit direct control of concentration and which typically unfold at slower timescales.

The jump to irreversible vs. reversible also belies the finesse of the approach that can measure sub-minute. There are serious biological implications to reversibility in <1 min vs. reversible repression in ~10 minutes. Recent genetic experiments have made it clear conserved enhancer (as for Scr) appear to function primarily to avoid these ~ minute reviews.

To me much of the discussion about irreversibility of repression hinges on our understanding of epigenetic inheritance, and I don't think most of the field would be surprised to see this cannot be stably (irreversibly) established within the scale of ~30 min time scales assayed here. What to me is more exciting is that the reversibility can be measured at all at these time scales. Other authors (e.g. Bintu...Elowitz 2016, Ref 13) have looked at the reversibility in mammalian (immortalized) cell culture at the time resolution of days, not minutes (not at the single molecule scale). Bintu has shown that the difference between 1 day and 5 days of induced repression have different degrees of reversibility. Thus, the big question is not so much the qualitative one of what is reversible vs. irreversible, but really knowing the quantitative time scales for these with resolutions of minutes, not days. We can do a much better job

testing/eliminating potential models of gene regulation with this revolutionary sub-minute temporal control introduced here.

Author response: This is an excellent point. We have revised the introduction in an effort to better convey the reviewer's point that the key point is not so much that we test the binary reversible/irreversible distinction, but instead the fact that we take high-resolution measurements that can quantify the timescale of repression and the rate at which (if at all) reversible repressor binding induces longer-lived repressed transcriptional states (lines 49-54).

2) Figure 2

I am confused by the numbers on the graphs on Fig 2D vs. Fig 2E. Is a different normalization being used? It appears the 3 examples in E all show transcription shutting down when the relative Kni levels reach 10 au (which continues to rise to 16 au). However, in 2D the max Kni levels are 10 au, and repression is at half max at 6 au (which is the level at which the graphs on 2E start). Similarly the transcription rates go from 0 to 20 in D and 0 to 1.2 in E. Either I'm reading this all wrong or there is some unnecessary discrepancy in the normalization. I appreciate it's all "arbitrary units" but it would be nice if the combined trace examples were on the same scale as the single cell examples.

Author response: We thank the reviewer for pointing out the discrepancies between Fig 2D and Fig 2E, and we sincerely apologize for any confusion that may have resulted from this inconsistency. Upon revisiting the figures in question, we uncovered an error in the fluorescence background subtraction process, which led to the discrepancies observed. We have promptly addressed and corrected this error, ensuring that the relevant figures, namely Figure 2A, Figure 2E, and Figure S8, are now revised to present accurate data. We hope that these corrections have resolved any confusion, and we thank the reviewer for bringing this to our attention.